# Optimization of Surface-Engineered Micropatterns on Bacterial Cellulose for Guided Scar-Free Skin Wound Healing

**DOI:** 10.3390/biom13050793

**Published:** 2023-05-05

**Authors:** Haiyan Liu, Yang Hu, Xiuping Wu, Rong Hu, Yingyu Liu

**Affiliations:** 1Shanxi Province Key Laboratory of Oral Diseases Prevention and New Materials, Shanxi Medical University School and Hospital of Stomatology, Taiyuan 030001, China; 2Center for Human Tissue and Organs Degeneration and Shenzhen Key Laboratory of Marine Biomedical Materials, Institute of Biomedicine and Biotechnology, Shenzhen Institutes of Advanced Technology, Chinese Academy of Sciences, Shenzhen 518055, China

**Keywords:** bacterial cellulose, lithography, micropatterns optimization, scar-free

## Abstract

Bacterial cellulose (BC) has been widely used in tissue engineering due to its unique spatial structure and suitable biological properties. In this study, a small biologically active Arginine-Glycine-Aspartic acid-Serine (RGDS) tetrapeptide was incorporated on the porous BC surface followed by a low-energy CO_2_ laser etching operation. As a result, different micropatterns were established on the BC surface with RGDS only anchored on the raised platform surface of the micropatterned BC (MPBC). Material characterization showed that all micropatterned structures exhibited platforms with a width of ~150 μm and grooves with a width of ~100 μm and a depth of ~300 μm, which displayed distinct hydrophilic and hydrophobic properties. The resulting RGDS-MPBC could hold the material integrity, as well as the microstructure morphology under a humid environment. In-vitro and in-vivo assays on cell migration, collagen deposition, and histological analysis revealed that micropatterns led to significant impacts on wound healing progress compared to the BC without surface-engineered micropatterns. Specifically, the basket-woven micropattern etched on the BC surface exhibited the optimal wound healing outcome with the presence of fewer macrophages and the least scar formation. This study further addresses the potential of adopting surface micropatterning strategies to promote skin wounds towards scar-free outcomes.

## 1. Introduction

Skin is the body’s static defense system, consisting of the epidermis, the dermis, and the vascularized connective tissue under the dermis [1]. The dermis layer is responsible for skin flexibility, strength, and retaining moisture to protect the body from infections and mechanical stress. Certain patterning information and positional identity are contained in the skin dermis, which may guide fibroblast cells to differentiate into various cell types that perform specific functions, such as hair follicle initiation and scar formation during wound tissue regeneration [2]. Specifically, when the dermis of the skin is damaged by pathological and mechanical stimulation, the nano/microscale topography and mechanical constraints imposed by the skin microenvironment would significantly impact the structure and functions of fibroblast cells, as well as various cellular behaviors, such as proliferation, migration, differentiation, gene expression, and cell-to-cell interaction [3,4,5,6,7]. For example, Reis and coworkers [3] found that the pattern on the hydrogels caused by the E-beam microlithography was able to adsorb proteins and precisely control the growth and organization of individual cells. Martinez and coworkers [8] reported that the topographic structures have direct effects on protein deposition due to the enhanced surface area caused by the nano-topographic surface structure. Regarding the tissue engineering, the change of the wound microenvironment caused by certain external factors, such as the use of surface micropatterning strategies, would impact the size, shape and biological behaviors of fibroblast cells, leading to the desired outcomes of cell proliferation, differentiation, and, furthermore, inducing the synthesis and deposition of related proteins towards good tissue repairs [9,10,11,12,13,14,15].

The wound scarring is a termination of wound healing progress. Severe and hardly healed wounds generally result in scars that impact not only the movement capability but also the aesthetics of beauty. Scientists have been dedicated to reducing the adverse effects of scar formation, as well as approaching the scarless wound healing outcome. Among them, our contribution was to use bacterial cellulose (BC) as a promising and cost-efficient nano-scaffold for wound tissue engineering [16,17]. Under the static culture using specific bacterial strains (*Gluconacetobacter xylinus*), the BC films may synthesize a pellicle-like hydrogen that has the same chemical composition as plant cellulose consisting of linear β-1, 4-d-glucose units [18]. In general, BC exhibits many unique properties, such as a high purity of cellulose, a high-water retention capacity, a high degree of crystallinity, in-vivo biocompatibility, a unique wet strength, and interconnected porosity, as well as multiple competencies of chemical and physical modifications [19,20]. In addition, the nanofibrous network of BC films with a fiber size of <100 nm leads to a higher surface area, which enables BC to have a stress-strain behavior resembling soft tissue. However, the BC nanostructure limits the specific applications of BC in tissue engineering because of a small pore size (100–300 nm) [21,22]. As a material to dress a wound, BC is an ideal option because its properties closely resemble the extracellular matrix of normal skin and provide a humid environment to promote the entire wound bed healing [23,24,25]. On the other hand, the positive biological behaviors of cells that can favorable guide wound healing progress are impacted by various factors, in which the pore size of biomaterials over the wound area plays an important role. Strategies have been developed to improve the pore size and modify the topography of BC in order to facilitate cell growth and protein distribution/deposition [26,27].

BC acts as a tissue engineering scaffold for wound healing with high tensile strength and flexibility [28]. However, due to the size of human dermal fibroblasts (HDF) (20–30 μm), the small pores of BC restrict its practical use for cell ingrowth against the formation of dense, parallel fibril bundles during wound healing [29,30]. One of our studies has exploited surface-patterning technology to solve the related wound healing issue to improve scar formation. The surface-patterning technology has been demonstrated as a viable approach to precisely manipulate cell migration and protein orientation in the ECM toward scar-free healing on the wound regeneration site [31,32]. It has been proved that the HDF cells seeded onto micropatterning-produced micro- and nano-grooves could align their shape and elongate in the direction of the groove [11]. Various techniques have been used to create different surface micropatterns, such as freeze drying [33,34], gas foaming [35], cryogelation [36], phase separation [37], and microfabrication techniques, including etching [38], micro-contact printing (µCP) [39,40,41,42], and photolithography [43,44]. It is well known that laser lithography can cause precise modifications to the material surfaces to generate contaminant-free surfaces without demolishing the inherent properties of the materials [45]. Our previous study revealed that low-energy carbon dioxide (CO_2_) laser (10 W) etching technology could efficiently generate raised platforms ~100 μm wide and grooves with ~150 μm wide on the BC surface [46]. The findings in this study preliminary demonstrate the potential of applying lithographical technology on the soft BC material surface to generate well-maintained micropatterns, capable of guiding cell migration and collagen deposition, and positively regulating scar-free wound healing.

Understanding the skin structure can assist in the design of precise micropatterns to guide the migration and distribution of cells, as well as the distribution of collagen fibers. In the dermis, less densely packed large-diameter collagen fibers form large basket-woven bundles of branching elastic fibers as a superstructure, and then elastic fibers form an elastic microfibrillar network with oxytalan and elaunin to be finally assembled into the reticular dermis [47,48,49,50]. As for the micropatterns on the material surface, changing micropatterns may directly or indirectly affect the remodeling and reconstitution of extracellular components [51]. To generate such a basket-woven pattern of fiber bundles on the BC surface to verify its efficacy of guiding the cell and collagen organization during the wound healing process, a low-energy CO_2_ laser was continuously used in this study, and the other two patterns (column and maze) were also used as contrasts. Therefore, three micropatterns (column, woven, and maze) were precisely established using computer-aid software following the tetrapeptide RGDS (Arg-Gly-Asp-Ser) attachment through the hydrogen bonding interaction [52]. The use of RGDS was hypothesized to help promote cellular motility, proliferation, and differentiation [53]. This study is the first endeavor to use micropatterns that mimic real collagen bundle patterns in the dermis on the BC material surface to investigate cell migration and collagen distribution for the scarless healing of skin wounds. Material characterization, including scanning electron microscopy, surface topographical measuring instrument, energy dispersive spectrometry, and the moisture test, as well as in-vitro and in-vivo biological assays, were performed to screen out the optimal micropattern. This study could advance the knowledge of real collagen bundle patterns in the dermis and further address the benefit of adopting the micropatterning strategies associated with collagen bundle patterns to promote wound healing for scar-free healing outcomes.

## 2. Materials and Methods

### 2.1. Materials

BC was purchased from Hainan Yida Food Co., Ltd. (Haikou, China), human dermal fibroblast cells (HDF, ATCC newborn 2310) were purchased from ScienCell Co., Ltd. (Beijing, China), and H-Arg-Gly-Asp-Ser-OH tetrapeptide (arginine-glycine-aspartic acid-serine, RGDS) was purchased from Wuhan MoonBiochem Co., Ltd. (Wuhan, China). Other chemicals, dyes, and cell growth media were purchased and noted elsewhere.

### 2.2. Preparation of Three Patterns of RGDS-MPBC

The detailed protocol for preparing the RGDS-anchored bacterial cellulose (RGDS-BC) was used as previously reported by our team [46]. Briefly, RGDS-BC were prepared using two steps, freeze-dried BC preparation and RGDS attachment. BC and RDGS-BC samples were freeze-dried and cut into 24-well-sized pieces. Three micropatterns (column, basket-woven, and maze) were generated under the multi-cycling etching process at a lower excitation power (10 W) using the CO_2_ Laser Cutting Instrument (CO_2_-D30-Y, Han’s Laser Co., Shenzhen, China) (Figure 1). To achieve approximate sizes for the widths of the platform (~150 μm) and the groove (~100 μm) on three patterns, the laser energy was controlled to increase incrementally. The RGDS could be left on the platform but be absent on the groove structure because of the laser etching-out operation. The prepared RGDS-MPBC materials were ready for material characterization.

### 2.3. Characterization of RGDS-MPBC

The surface topographical measuring instrument (STMI, HSP-LI, Red Star Yang Technology, Wuhan, China) was used to observe the surface topographical structure. Prior to observation, the samples were freeze-dried and graphed using a low-energy laser and then a flat surface was selected. The field emission scanning electron microscope (FESEM, FEI Nova NanoSEM 450, Eindhoven, The Netherland) was used to observe the morphology of three different micropatterns, and the energy dispersive spectrometer (EDS, Nova NanoSEM 450, Eindhoven, The Netherland) was used to evaluate the anchored small molecule bioactive tetrapeptide on the raised platforms of RGDS-MPBC. For FESEM observation and EDS analysis, the small pieces of samples were attached to the carbon tape and then coated with gold for 30 s under a vacuum.

### 2.4. Moisture Absorption Test

To determine the moisture absorption effect, the freeze-dried BC, MPBC, and RGDS-MPBC samples (2 cm × 2.5 cm) were placed under 75% RH (relative humidity) at 25 °C. Prior to the absorption experiments, all samples were attained at constant weight after being vacuum dried. After certain intervals (2, 4, 8, 12, 24, 36, 48, and 96 h), all samples were taken out and weighed on balance (PR124ZH, with a precision 0.1 mg). The moisture uptake result at any time was determined by:$$Moisture\;uptake=\frac{W_{h}-W_{0}}{W_{0}}$$
where W_h_ and W_0_ were the weights of humid samples after the moisture absorption and dry samples without exposure to moisture absorption. All data were statistically analyzed from at least three repeated tests.

### 2.5. In-Vitro Study of RGDS-MPBC

The prepared RGDS-MPBC materials were cut into pieces that fit a 24-well tissue culture plate. High-energy (25 kGy) electron acceleration irradiation was used to sterilize all the samples for 6–7 h; the electron beam is safe enough for biomolecules [54]. Sterilized RGDS-MPBC pieces were placed into the tissue culture plate and 2 mL of phosphate buffered saline (PBS) was added. On day 2, the excess PBS solution was removed and the specific fibroblast growth medium (FGM; ScienCell Research Laboratories, Carlsbad, CA, USA) was added to soak the pieces for one day. The HDF cell suspension with an initial density of 1 × 10^5^ cells/mL per well was then added into the tissue culture plates. The cells were cultured at 37 °C with 5% CO_2_ and the FGM with fetal bovine serum (FBS) was replaced every 2 days. To begin the in-vitro study, on days 3, 7 and 10, 1 mL of PBS was added to rinse the sample pieces and then 4% paraformaldehyde solution was used to fix the sample at room temperature for 30 min. After that, the samples were stained with 10 nmol Actin-stain^TM^ 488 Fluorescent Phalloidin (Cytoskeleton Inc., Denver, CO, USA) for 30 min and then 20 mg/mL of 4,6-diamidino-2-pbenylindole (DAPI, Dojindo, Kumamoto, Japan) fluorescence dye for 30 s at room temperature. The cellular behavior of stained HDF cells, including attachment, growth, and proliferation, was observed on a fluorescence microscope (Olympus, Tokyo, Japan).

### 2.6. In-Vivo Animal Study

The in-vivo animal study was approved by the Shenzhen Institute of Advanced Technologies at the Chinese Academy of Sciences with the approval permit code SYXK (Guangdong) 2012–0119. The 32 Sprague Dawley (SD) rats weighing between 200 and 250 g (Beijing Vital River Laboratory Animal Technology Co., Ltd., Beijing, China) were divided into four groups, and a pairwise-control experimental design was performed. All SD rats were kept in a sterile environment and regularly supplemented with sterile food (Beijing Keao XieLi Feed Co., Ltd., Beijing, China) and water. All the materials containing pristine BC and RGDS-MPBC with three kinds of micropatterns were soaked in sterile saline containing 0.2% Poly (iminocarbonimidoyliminocarbonimidoylimino-1,6-hexanediyl) hydrochloride (PHMB) to enhance the anti-inflammatory effect. To start the experiments, each SD rat was injected with sodium pentobarbital of 30 mg/kg rat weight. The skin (5 × 5 cm) on either side of the spine was then removed to generate a full-thickness wound (1 × 2 cm) with a thickness of 6 mm using a sharp sterile surgical scalpel. The wound area was cleaned and disinfected with 10% povidone-iodine, and subsequently fully covered with various sterile BC materials. Finally, the wound areas and the sample materials above the wound areas were covered with sterile gauze and closed with a sterile suture (Shenzhen RWD, Shenzhen, China).

On day 7, 14, 21, and 28, the SD rats were anesthetized and a full-thickness skin section (complete epidermis and dermis) was excised and fixed in 4% paraformaldehyde solution (Aladdin F111934, Germering, Germany) before storing at 4 °C for 12 h. The fixed tissue was rinsed and then dehydrated in different concentrations of ethanol: 50% and 70% for 30 min, 80% for 20 min, 90% for 20 min, and then anhydrous ethanol for 30 min, twice. The paraffin-embedded (Leica Paraplast, Teaneck, NJ, USA) dehydrated tissue using an embedding machine (KL-1, Mingke Technology, Guangzhou, China), and then a microtome (Leica RM2235 Manual Rotary Microtome, Teaneck, NJ, USA) was used to make sample tissue slices with a thickness of 5 μm. Tissue slices were then dried at 60 °C for 1 h and stained with a hematoxylin and eosin (H&E) protocol, tissue slices were observed under the upright microscope (Olympus BX53, Hamburg, Germany) to evaluate wound healing progress and outcomes for each BC material.

## 3. Results and Discussion

### 3.1. Characterization of Different Micropatterns on RGDS-MPBC

As shown in Figure 2a, the BC films harvested from the medium show a pellicle-like hydrogel formation. Small pores (100–300 nm) are predominantly present in the network structure of BC film as shown in Figure 2b,c. In order to facilitate cell growth and protein distribution/deposition, we continued to adopt low-energy CO_2_ laser photolithography following the RGDS attachment on the BC film to generate different micropatterns that would guide the cell migration and collagen deposition for better wound healing towards scar-free healing outcomes. In Figure 3, three micropatterns (column, basket-woven, and maze) were established on the BC surface with precisely controlled shapes of grooves and platforms. Under this scenario, a series of microstructures on the BC were generated that may influence versatile cellular behaviors during the wound healing process. SEM and 3D STMI images in Figure 3a–c exhibit explicit surface morphologies associated with different micropatterns, suggesting the success of applying the low-energy CO_2_ laser photolithography on the BC film. Despite the different shapes of platforms (square, rectangular, and curved rectangular), the width of the platforms between the grooves for all three micropatterns is ~100 μm and the width of the groove is ~150 μm, as shown in Figure 3a–c. The groove shape is similar to its platform shape, and the groove depth from the platform surface for each micropattern is estimated to be ~300 μm as shown in Figure 3A–C.

BC is a highly hydrophilic biomaterial. Our previous studies have demonstrated that the chemical structure of cellulose determines the surface electronegativity and the application of micropatterning technology can enhance its surface hydrophobicity to overcome the inadequate affinity to the collagen [46,55]. Furthermore, the RGDS tetrapeptide anchored on the BC surface can improve cell attachment through the hydrogen bonding between the side groups of RGDS and the free hydroxyl groups of cellulose [56]. In this study, the BC was first soaked in the RGDS solution to allow the RGDS to be anchored onto the BC surface to benefit the cell attachment. Afterwards, the CO_2_ laser lithography was executed to burn out RGDS in the grooves while still allowing RGDS to remain on the raised platform surfaces. EDS scanning results as shown in Figure 4a-i,b-i,c-i, exhibit the presence of nitrogen (N) on the surface of the RGDS-MPBC platform, which is proof of RGDS residue. Meanwhile, no N element was detectable in the groove channels (in Figure 4a-ii,b-ii,c-ii), suggesting that RGDS was burned out. The collagen prefers to be adsorbed on the hydrophobic surface, and a hydrophilic surface is favorable for fibroblast cell attachment. Under this scenario, dual affinities to fibroblast cells and collagen on the platform surfaces and in the grooves, respectively, were generated on the RGDS-MPBC materials.

In general, the dressing should play an important role in preventing bacterial invasion at the wound interface, permitting gaseous exchange, and offering a benign microenvironment for wound healing [57]. Other than these above advantages as a biomaterial applied to a complex wound environment, bacterial cellulose should exhibit significant biocompatibility [58]. The establishment of the micropattern on the BC surface could have enhanced its surface roughness, which could significantly reduce the moisture absorption due to the air being partly filled in the gap because of the micropatterns than that of a flat surface [59]. Therefore, all the different surface micropatterns can increase the hydrophobicity on the BC surface. Figure 5 shows the dynamic moisture test on different BC surfaces with and without RGDS and micropatterns within 60 h. It can be observed that the RGDS-MPBC exhibits better moisture than pure BC, which may effectively prevent the production of wound pus and secretions [24].

### 3.2. In-Vitro Cell Study

To further study cell migration that may potentially mediate collagen distribution on the BC with different micropatterned structures, HDF cells were cultivated on BC and RGDS-MPBC. Figure 6 shows that different BC surfaces with and without micropatterns and RGDS-targeted attachment manipulate HDF cell distribution and proliferation over time. The effect of micropatterns on manipulating the migration of HDF cells is significant in all three RGDS-MPBC samples when compared to the untreated BC. As the cultivation continued, HDF cells (blue images with DAPI stained) were gradually concentrated from the groove section to the platform surface for the RGDS-MPBC samples. In addition, the cell density was also gradually enhanced as the HDF cells proliferation progressed on the platform surface from day 1 to day 7, which has been illustrated by the cell proliferation from day 3 to day 7 in Figure 6. For the collagen distribution observation, since the collagen (green images with 488 Fluorescent Phalloidin) is mostly from cell secretion [60], it is still mainly present on the platform surface on RGDS-MPBC samples from day 3 to day 7. However, after the HDF cell started being migrated, some of the collagen emerged on the edge of the platform surface and gradually filled the grooves from day 7 to day 10. The woven micropattern seems to have more densely filled collagen in the groove section than other micropatterns, suggesting that the woven micropattern may have a better dual affinity to HDF cells and collagen to guide their migration and organization. The preliminary in-vitro study may demonstrate the efficacy of different micropatterns manipulating cell migration and collagen distribution. In terms of the real skin collagen organization that mimics a basket-woven bundle structure, the woven micropattern may be the most suitable microstructure that could be constructed on the BC surface to benefit scar-free wound healing.

### 3.3. Evaluation of the RGDS-MPBC in the In-Vivo Assay

SD rats were used for the in-vivo study to preliminarily evaluate wound healing towards scar-free outcomes using various micropatterned BC samples. Our previous study demonstrated the scar-free healing potential of the micropatterned BC materials [46]. The wound tissues covered by surface-micropatterned materials made of RGDS anchoring plus crossed column-groove micropatterns did not undergo deformation, degeneration, and necrosis, while the newborn collagen fibril bundles seemed to be relatively thick and were distributed with a low density in the formed scar [61]. In this study, two new micropatterns, woven and maze, were compared with the column micropatterns to screen out the optimal micropattern that may lead to the healing outcome with the least scarred tissue organization. As shown in Figure 7, before the wound was completely closed, a significant inflammatory reaction occurred around the wound area, including a small amount of pus. This is due to the body’s immune response to foreign substances. On day 21, these responses began to fade away. For example, Figure 7a-1–a-4 shows a trend that the wound area was gradually reduced over time. Comparably, the wounds covered by all RGDS-MPBC samples exhibit a better healing outcome, and the wound covered by RGDS-MPBC-woven was closed more toward the center of the wound and it seems to have a smaller closed scar on the skin compared to the other two micropatterns. This suggests that various micropatterns may have caused different impacts on the outcome of wound healing and the woven micropatterns seem to have less scar formation. Furthermore, Figure 8 shows the histological analysis of newly regenerated tissues in the wound area on day 28. The results observed in this study are similar to what we have found in our previous study [46], showing that collagen fibril bundles and newborn capillary vessels are more evenly distributed and denser in the wounds covered by three micropattern samples than in the original BC sample. Meanwhile, it was noticed that the formed collagen fibril bundles on the wound covered by RGDS-MPBC-woven and RGDS-MPBC-maze samples were thinner and more uniform than the RGDS-MPBC-column sample. In particular, the RGDS-MPBC-woven sample exhibits a formation of fewer macrophages on day 28 than all the other samples, suggesting less inflammation occurring with potentially less scar tissue formation. The above in-vivo animal study preliminarily demonstrated the positive impact of various micropatterns on wound healing towards the scar-free healing outcome. Further study needs to be conducted with a comprehensive study on multiple intrinsic growth factors and wound healing signaling pathways related to the application of different micropattern samples that may cause different wound healing efficacies.

## 4. Conclusions

Lithographically patterning technology has been well documented to be a superior strategy to engineer the surface of biomaterials in order to exhibit advanced properties for benefiting cell-oriented growth and protein adsorption in tissue engineering. BC is a promising biomaterial, and imparting BC as a stimuli-responsive smart feature may advance its profound applications in tissue engineering. In this study, we continued to investigate the surface micropatterns created using a low-energy CO_2_ laser treatment on the BC, specifically focusing on the optimization of various micropatterns (column, woven, and maze) for guided scar-free skin wound healing. Firstly, the low-energy CO_2_ laser treatment was able to control the precise micropatterns to distinguish various microstructures on the BC surface, and it was also able to ensure the integrity of the groove section without penetrating the material, as well as the integrity of the whole micropattern structures in the hydrated state. Regardless of the various shapes of micropatterning platforms (square, rectangular, and curved rectangular), the width of the platforms between grooves for all three micropatterns is ~100 μm and the width of the groove is ~150 μm, and the groove depth is ~300 μm with a similar shape to its platform surface. The targeted RGDS tetrapeptide attachment followed by the CO_2_ laser treatment led to a dual affinity to the cells and the collagen that has been demonstrated by EDS scanning results, moisture testing results, and in-vitro cell assay. The RGDS exclusively present on the platform attracted cell migration, which has been proved using the cell fluorescence staining assay. The micropatterning groove area was gradually filled by the collagen fibrils secreted by the HDF cells, which has been also demonstrated by the cell fluorescence assay. As compared to other micropattern samples, the RGDS-MPBC-woven sample, mimicking the basket-woven collagen organization of the real skin tissue, exhibited a better wound healing outcome with a smaller wound close, less scarred tissue, thinner/denser/more uniform collagen fibril bundles, and the presence of fewer microphages in the in-vivo assay. Such a specific woven-design micropatterning strategy could have potential in employing advanced surface engineering micropatterning technology to achieve a scar-free wound healing outcome.

## Figures and Tables

**Figure 1 biomolecules-13-00793-f001:**
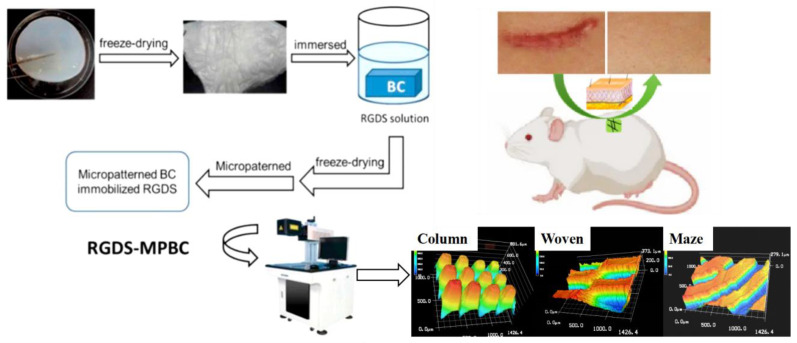
Schematic illustration of the RGDS-MPBC preparation, 3D surface images of different micropatterns, and in-vivo study process.

**Figure 2 biomolecules-13-00793-f002:**
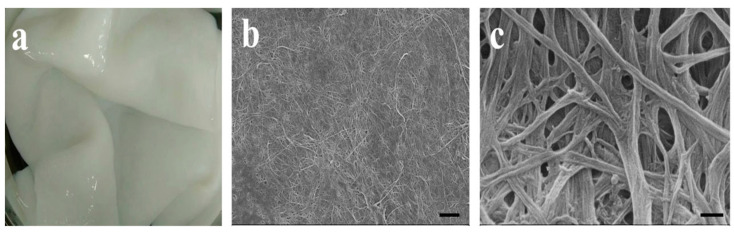
Photographs of (**a**) BC hydrogel material; FESEM images of (**b**) surface and (**c**) porous network structure of BC hydrogel material. Scale bar in (**b**): 1 μm. Scale bar in (**c**): 200 μm.

**Figure 3 biomolecules-13-00793-f003:**
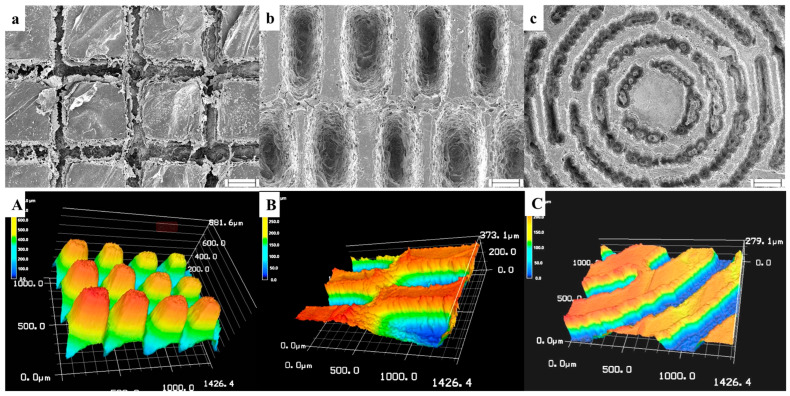
(**a**–**c**) FESEM images and (**A**–**C**) 3D STMI observation images of RGDS-MPBC with three micropatterns. Scale bar in (**a**–**c**): 50 μm.

**Figure 4 biomolecules-13-00793-f004:**
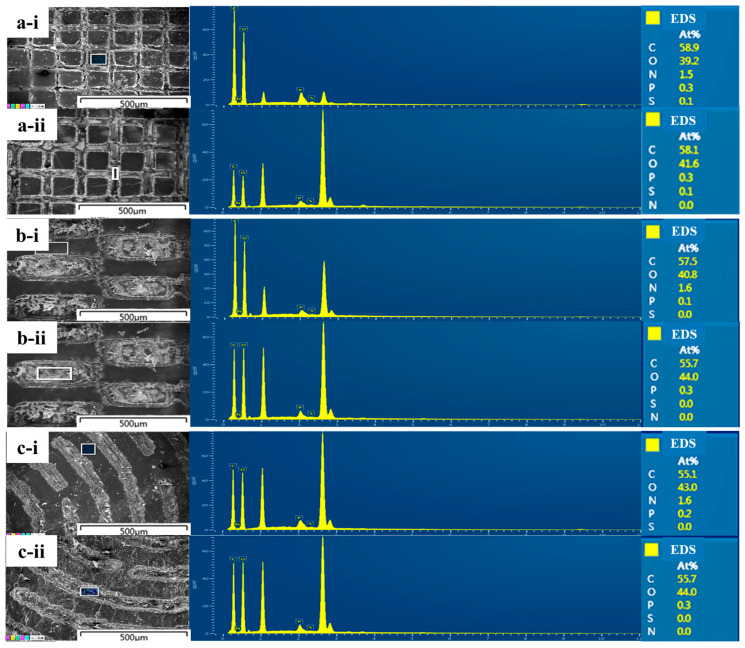
EDS results (**right**) from the square symbols as shown in FESEM images (**left**) of the RGDS-MPBC with different micropatterns for platforms and grooves in (**a-i**) and (**a-ii**) column, (**b-i**) and (**b-ii**) woven, (**c-i**) and (**c-ii**) maze, respectively.

**Figure 5 biomolecules-13-00793-f005:**
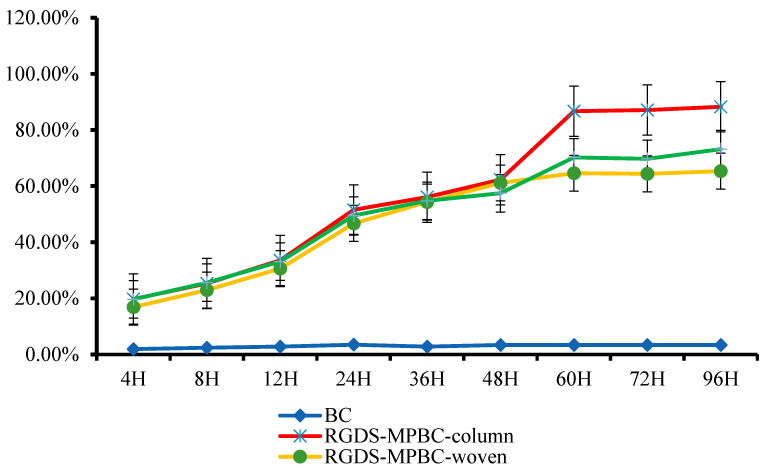
The moisture test results of BC, RGDS-MPBC-column, RGDS-MPBC-woven, and RGDS-MPBC-maze.

**Figure 6 biomolecules-13-00793-f006:**
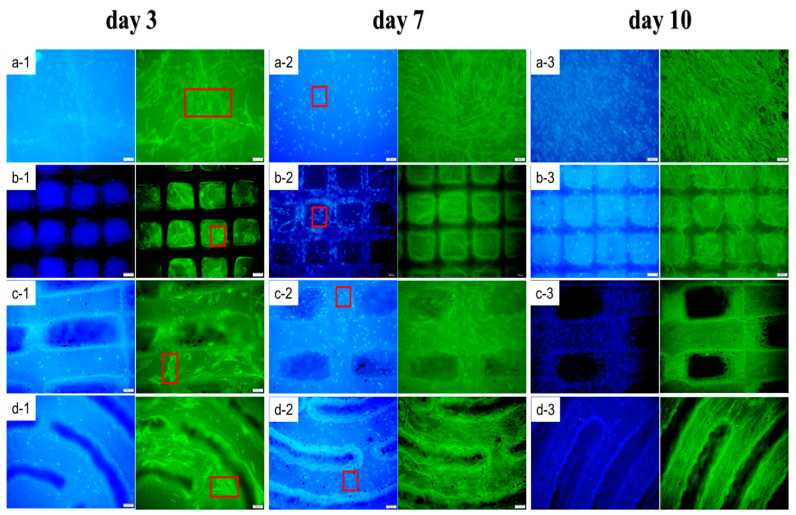
Growth and migration of HDF cells on (**a**) BC and (**b**–**d**) RGDS-MPBC on day 3, 7, and 10. Cells (red square) were stained using DAPI and Actin-stain^TM^ 488 Fluorescent Phalloidin and observed under fluorescence microscope. Scale bar in each figure is 100 μm.

**Figure 7 biomolecules-13-00793-f007:**
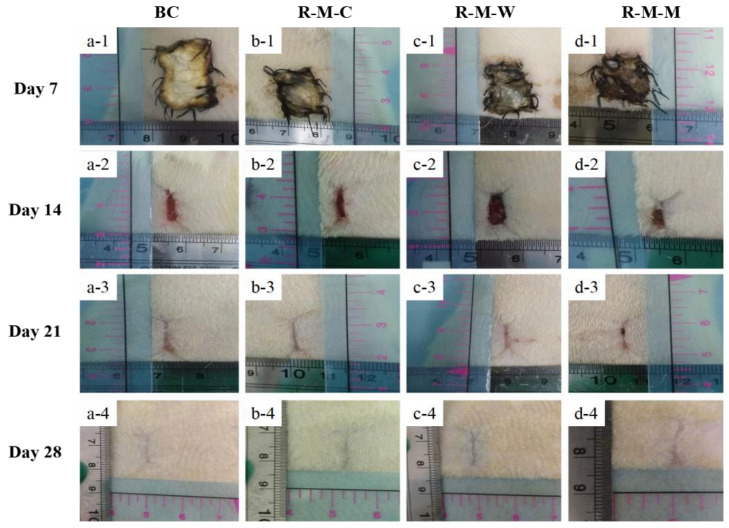
Examples of representative photographic processes illustrating skin tissue regeneration under BC and different micropatterning RGDS-MPBC during a period of 4 weeks: (**a-1**–**a-4**) BC as a contrast; (**b-1**–**b-4**) RGDS-MPBC-column; (**c-1**–**c-4**) RGDS-MPBC-woven; and (**d-1**–**d-4**) RGDS-MPBC-maze.

**Figure 8 biomolecules-13-00793-f008:**
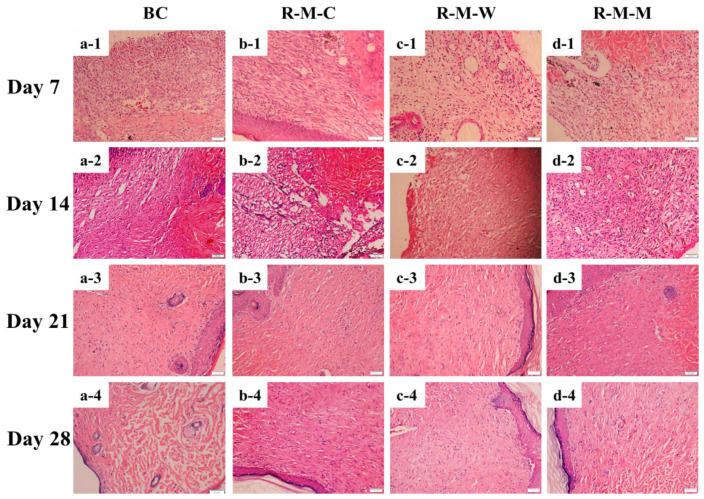
Examples of histological images of regenerated skin tissue on the healing stage for 4 weeks: (**a-1**–**a-4**) BC as a contrast; (**b-1**–**b-4**) RGDS-MPBC-column; (**c-1**–**c-4**) RGDS-MPBC-woven; and (**d-1**–**d-4**) RGDS-MPBC-maze. Scale bar in each figure is 50 µm.

## Data Availability

The data presented in this study are available on request from the corresponding authors. The data are not publicly available due to privacy and intellectual property issues.

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
