# Peer review of "Optimization of Surface-Engineered Micropatterns on Bacterial Cellulose for Guided Scar-Free Skin Wound Healing"

_biomolecules, 2023, doi:10.3390/biom13050793_

Round 1

Reviewer 1 Report

In section 3.1 the authors claim "In this study, the BC was first soaked in the RGDS solution to allow the RGDS to be immobilized onto the BC surface to benefit the cell attachment", how are they being immobilised onto the surface of the BC? Is there a protein or chemical modification being made to the BC surface or is it just an incubation of the BC in a solution of the RGDS and thus incorporation? This may be an issue when it comes to incubation in liquid media as it would allow for diffusion and hence less spatial control over the positioning of the RGDS. I would change the word immobilised if there is no chemical immobilisation taking place. 

In regards to Figure 6. All images need to be made clearer and enlarged as it is very hard to determine any cell proliferation among the background noise. Increase the contrast, make the images larger and clearer. If we are looking at a section of the micropattern it may be easier to concentrate on one section of the pattern which shows the difference between patterned cell proliferation and non-patterned. It is hard to make out what the arrows are pointing to due to the size of the images and the quality as they are quite blurry therefore please use a bordered square in order to focus on the section of the image to be emphasised. 

Also in section 3.1, the authors refer to "day 1" yet there is no image of day 1 in figure 6? Please address this if possible. 

Figure 7 also has no day 1 image, please address this if possible. 

Figures 6,7 and 8 are highly qualitative instead of quantitative, there should be quantification of fluorescence or difference of fluorescence compared to control in square sections in order to draw the conclusions in figure 6 (however, it may be the case of the of image quality and size so please see if this addresses the case first). Similarly figure 7, as there is no quantification, seems to show that there is very similar results across all samples. Are these averages of the scar result? As we cannot determine any effect if there is no replicate data on the scar formation. 

An interesting paper with some tweaks will make for a great read! Thank you to the authors. 

Reviewer 2 Report

The Authors aimed at developing micropatterns to mimic the real collagen bundle patterns in the dermis on BC material surface to investigate cell migration and collagen distribution towards scarless healing of skin wounds. Despite the fact this reviewer is not an English native speaker, the spelling is good and clear. The authors have successfully reached the proposed goal. To do so, they have used characterization techniques that corroborates all their findings. The proposed micropatterns showed a clear influence on the tissue healing, as seen in the in-vivo analysis. The results are of great importance to tissue engineering, and introduces new data to this field. Taking into account the aforementioned, this reviewer suggests acceptance of this work as it is. 

Hoppe the best.

Reviewer 3 Report

The authors report on the construction of micropatterns on bacterial cellulose (BC), previously functionalized with a biologically active peptide (RGDS). The micropatterns are obtained by using low-energy photo-lithography.  The material so obtained shows the presence of platforms and grooves with distinct hydrophilic and hydrophobic character, which can guide the human dermal fibroblasts (HDF) distribution and proliferation over time. The final aim is to obtain a dispositive improving wound healing and limiting scar formation. Various micropatterns are produced and characterized also by in -vitro and in-vivo assays on cell migration, collagen deposition, and histological analysis. All the tested micropatterns improve the wound healing processes compared to pure BC, with an optimal wound healing outcome with the presence of fewer macrophages and minor scar formation for the basket-woven micropattern.

 The basic idea of this paper was the object of a former paper (Material Science of Engineering c, 99, 2019, 333) by the same authors. Moreover:

 the text is confusing, the first paragraph of the results is an introduction,

the references are often inappropriate, and the relevant literature in the field is ignored,

the images have a low resolution (see for example the fluorescence images of Figure 6 where the cells are hardly distinguishable)

so that it is difficult to appreciate any element of novelty, if present.

For these reasons, I suggest discarding the paper in its present form.

Round 2

Reviewer 3 Report

The authors improved the manuscript as suggested.